# Effects of Nifedipine on Renal and Cardiovascular Responses to Neuropeptide Y in Anesthetized Rats

**DOI:** 10.3390/molecules26154460

**Published:** 2021-07-24

**Authors:** Angela Bischoff, Martina Stickan-Verfürth, Martin C. Michel

**Affiliations:** 1Arensia Exploratory Medicine GmbH, 20225 Düsseldorf, Germany; angela.bischoff@arensia-em.com; 2Department of Nephrology and of Particle Therapy, University Hospital Essen, West German Proton Therapy Centre, 45147 Essen, Germany; martina.stickan-verfuerth@uk-essen.de; 3Department of Pharmacology, Johannes Gutenberg University, 55131 Mainz, Germany

**Keywords:** neuropeptide Y, Y_1_ receptor, Y_5_ receptor, nifedipine, blood pressure, renal blood flow, diuresis, natriuresis

## Abstract

Neuropeptide Y (NPY) acts via multiple receptor subtypes termed Y_1_, Y_2_ and Y_5_. While Y_1_ receptor-mediated effects, e.g., in the vasculature, are often sensitive to inhibitors of L-type Ca^2+^ channels such as nifedipine, little is known about the role of such channels in Y_5_-mediated effects such as diuresis and natriuresis. Therefore, we explored whether nifedipine affects NPY-induced diuresis and natriuresis. After pre-treatment with nifedipine or vehicle, anesthetized rats received infusions or bolus injections of NPY. Infusion NPY (1 µg/kg/min) increased diuresis and natriuresis, and this was attenuated by intraperitoneal injection of nifedipine (3 µg/kg). Concomitant decreases in heart rate and reductions of renal blood flow were not attenuated by nifedipine. Bolus injections of NPY (0.3, 1, 3, 10 and 30 μg/kg) dose-dependently increased mean arterial pressure and renovascular vascular resistance; only the higher dose of nifedipine (100 μg/kg/min i.v.) moderately inhibited these effects. We conclude that Y_5_-mediated diuresis and natriuresis are more sensitive to inhibition by nifedipine than Y_1_-mediated renovascular effects. Whether this reflects a general sensitivity of Y_5_ receptor-mediated responses or is specific for diuresis and natriuresis remains to be investigated.

## 1. Introduction

Neuropeptide Y (NPY) is a co-transmitter of sympathetic neurons and adrenomedullary cells and contributes to the regulation of cardiovascular [1,2] and renal function [3]. NPY elicits its physiological effects via a family of G protein-coupled receptor subtypes named Y_1_, Y_2_ and Y_5_ [4]. Typical effects mediated by Y_1_ receptors include vasoconstriction in several vascular beds [2] including the mesenteric [5,6,7], renal [8,9,10,11], pulmonary [12] and cutaneous perfusion [13]. Y_1_ receptors can also mediate the potentiation of vasoconstriction by other vasoactive agents [14,15]. Y_2_ receptors have primarily been studied as prejunctional inhibitors of neurotransmitter release, both on sympathetic [16,17] and parasympathetic nerves [18] and in the brain [19]. The Y_5_ receptor has largely been studied in the brain where it is involved in the regulation of food intake [20], but it also contributes to the regulation of renal function and glucose metabolism [21].

The canonical signaling pathway of NPY receptors involves pertussis toxin-sensitive G proteins of the G_i/o_ family leading to inhibition of adenylyl cyclase [4]. Additional signal transduction pathways of NPY receptors, mostly restricted to specific cell types, include activation of phospholipases A_2_, C and D and modulation of K^+^ channel activity. Moreover, stimulation of NPY receptors often leads to elevation of the free intracellular Ca^2+^ concentration, which occurs partly secondary to the activation of phospholipase C and partly by inositol phosphate-independent mobilization of Ca^2+^ from intracellular stores [22,23,24] and by the influx of extracellular Ca^2+^ via L-type channels [6,7,25,26]. The latter is often probed by using inhibitors of such channels such as nifedipine [6,15,27,28,29]. However, in contrast to Y_1_ receptors, Y_2_ receptor stimulation can inhibit voltage-dependent Ca^2+^ channels [30,31]. Most research on the signal transduction of NPY receptors has been based on models of Y_1_ receptors and much less is known about signal transduction via other subtypes, particularly Y_5_ receptors [32]. Against this background, the present studies have used the L-type Ca^2+^ channel inhibitor nifedipine to probe the involvement of such channels in the regulation of diuresis and natriuresis representing Y_5_ receptors [33] as compared to mean arterial pressure (MAP), renal blood flow (RBF) and renovascular resistance (RVR) representing Y_1_ receptors [33].

## 2. Results

### 2.1. Study I (NPY Infusion)

Injection of nifedipine (3 µg/kg intraperitoneally) had little effect on basal values of most hemodynamic parameters except for an elevation of heart rate (HR; mean difference with 95% confidence interval: 47 [21; 73]), but increased urine output by 0.177 [0.035; 0.319] µL/15 min and natriuresis by 0.8 [0.38; 1.22] µmol/15 min (Table 1).

Infusion of NPY caused minor if any changes of MAP but rapidly reduced HR by about 30 bpm and RBF by about 3 mL/min with corresponding increases in RVR. These changes partly recovered during the 60 min observation period (Figure 1). Administration of nifedipine had only minor if any effects relative to its vehicle for any of these four parameters in the 60 min following the start of the infusion of vehicle (Figure 1). Infusion of NPY time-dependently increased diuresis and natriuresis with little effect on creatinine clearance (Figure 2). Nifedipine alone had only minor if any effects on any of these three parameters in the 60 min following the start of the infusion of vehicle but attenuated the NPY-induced diuresis and even more so natriuresis (Figure 2). For instance, the mean difference of urine and sodium output in the last 15 min of the NPY infusion in the absence and presence of nifedipine was 64 [1; 126] µL/15 min and 6.3 [0.9; 11.7] µmol/15 min, respectively.

### 2.2. Study II (NPY Bolus Injection)

Infusion of nifedipine (30 and 100 μg/kg/min) numerically reduced basal HR; however, CI of differences vs. vehicle included 0 and there was no obvious dose–response relationship (Table 2). Nifedipine had no major effects on basal MAP and RBF. Bolus injections of NPY (0.3, 1, 3, 10 and 30 μg/kg) caused dose-dependent elevations of MAP and reductions of HR and RBF; correspondingly, RVR increased (Figure 3). Nifedipine did not affect reductions of HR and had only minor effects on any of the other parameters in the 30 μg/kg/min dose (Figure 3). However, some attenuations of NPY-induced alterations were observed for MAP, RBF and RVR at the 100 μg/kg/min dose (Figure 3).

## 3. Discussion

Infusion or bolus injection of NPY moderately increases MAP [8,10,33,34,35,36,37], lowers HR [34], lowers RBF and increases RVR [8,10,11,33,34,36,37,38]. While most agents lowering RBF also decrease diuresis and natriuresis [3], infusion of NPY markedly increases diuresis and natriuresis [8,33,34,35,36,37] but does not affect CCR [8,36]. These findings were confirmed in the present study. Studies based on subtype-selective peptide analogs of NPY and on the Y_1_-selective non-peptide antagonist BIBP 3226 have established that reductions of RBF occur via Y_1_ receptors [11,28,33,38], whereas enhancements of diuresis and natriuresis occur via Y_5_ receptors [33]. The renovascular and tubular effects of NPY are also discriminated by the NPY receptor antagonist D-myo-inositol 1,2,6-triphosphate [8] and the cyclooxygenase inhibitor indomethacin [36], with the former mainly inhibiting the renovascular and the latter mainly the tubular effects of NPY. Studies with losartan had shown that the attenuation of the renovascular effects over time involves AT_1_ angiotensin receptors, whereas the diuretic and natriuretic responses do not [37]. The involvement of different receptor subtypes and mediators may explain why NPY increases diuresis and natriuresis despite lowering RBF.

Responses proven or at least bona fide assumed to be Y_1_ receptor-mediated are often sensitive to inhibitors of L-type Ca^2+^ channels, particularly in blood vessels [6,25,26,31]. As we are not aware of studies on the involvement of L-type Ca^2+^ channels in responses mediated by Y_5_ receptors, the present studies were designed to explore this for diuretic and natriuretic responses to NPY. Effects on RBF as model of Y_1_ receptors were explored in comparison.

We found that nifedipine more than doubles basal urine excretion, which is in line with the diuretic effects of nifedipine and other Ca^2+^ channel inhibitors in humans [39]. In our studies based on infusion of NPY, treatment with nifedipine attenuated the diuretic any natriuretic responses, indicating that Y_5_ receptors may also use L-type Ca^2+^ channels as a signal transduction pathway. As Y_5_ receptors were proposed to exhibit a restricted coupling to signal transduction pathways other than inhibition of adenylyl cyclase [32], these data support the idea that at least tubular Y_5_ receptors use additional signal transduction pathways. Given that indomethacin had inhibited tubular but not renovascular NPY responses [36], the present data do not allow definitive conclusions as to whether the involvement of L-type Ca^2+^ channels reflects a direct coupling of Y_5_ receptors; alternatively, it may reflect that receptors activated by cyclooxygenase products may lead to activation of such channels.

Several studies have demonstrated involvement of L-type Ca^2+^ channels in NPY-induced vasoconstriction, mostly found to involve Y_1_ receptors [6,25,26,31] and renovascular NPY responses are mediated by Y_1_ receptors [11,28,33,38]. Therefore, we were surprised to observe that the intraperitoneal injection of nifedipine had little effect on reductions of RBF induced by infusion of NPY. However, interpretation of these findings is complicated because RBF reductions by infusion of NPY are transient, and because the lack of effect of a given dose of inhibitor does not necessarily rule out involvement of a mechanism, as under-dosing may have occurred. Moreover, the infusion design does not easily allow testing of multiple NPY doses within an animal. Therefore, we performed a second study to address these shortcomings by using bolus injections of NPY and continuous infusion of two doses of nifedipine. The lower dose of nifedipine caused little inhibition of NPY-induced reductions of RBF, whereas the higher dose of nifedipine shifted the concentration–response curve of NPY. These data support the idea that the lack of effect of nifedipine on RBF reductions caused by infusion of NPY may at least partly reflect under-dosing. As the same dose of nifedipine had inhibited NPY-induced diuresis and natriuresis, it may be possible that tubular NPY effects are more sensitive to nifedipine than vascular effects. In a similar vein, it had been observed that nifedipine can inhibit urinary bladder contractions induced by a muscarinic receptor agonist in concentrations that are lower than those required to inhibit vascular smooth muscle contraction [40,41].

To the best of our knowledge, the present data provide the first evidence that Y_5_ NPY receptor-mediated effects of diuresis and natriuresis are attenuated by the L-type Ca^2+^ channel inhibitor nifedipine. Of note, this occurred at an exposure level to nifedipine where Y_1_-mediated vascular effects were not affected to a major extent. Given that Y_5_ receptors apparently exhibit a restricted coupling to signal transduction mechanisms other than inhibition of adenylyl cyclase [32] and that the tubular NPY responses appear to occur secondary to formation of cyclooxygenase activation [36], we cannot determine from the present data whether Y_5_ receptors couple directly to L-type Ca^2+^ channels. Moreover, we cannot determine whether the sensitivity to nifedipine is specific for tubular NPY responses or also applies to other effects mediated by Y_5_ receptors stimulation such as stimulation of food intake [20] or modulation of glucose homeostasis [21]. Additional in vitro studies, preferentially testing Y_5_ receptors expressed in multiple cell types, will be required to address this question.

## 4. Materials and Methods

The experiments had been approved by the state animal welfare board at the Regie-rungspräsident Düsseldorf).

### 4.1. Study I (NPY Infusion)

Study I used previously reported methods [33,34] with minor modifications: Briefly, 38 male Wistar rats (strain Hsd/Cpb:WU, weight 298–415 g, Harlan CPB, Zeist, The Netherlands) were unilaterally nephrectomized (left kidney) 7–10 days prior to the experiment under ketamine/xylazine anesthesia (100 and 6 mg/kg, respectively). On the day of the experiment, the rats were anesthetized with an initial intraperitoneal injection of sodium pentobarbitone (60 mg/kg), and additional doses of 3 mg i.v. were administered every 30 min. Body temperature was maintained at 37 °C by placing the rats on a heating pad and monitored via a rectal thermometer. Following tracheotomy to facilitate ventilation, the left femoral artery and vein were cannulated for monitoring MAP and HR (via a Statham pressure transducer) and for infusion of vehicle and NPY solutions, respectively. Following an abdominal midline incision, the connective tissue was carefully dissected from the right renal artery, and electromagnetic blood flow sensors (Skalar MDL 1401; Föhr Medical Instruments, Seeheim-Ober Beerbach, Germany) were placed on the vessels for monitoring RBF. The signals from the pressure transducer and the flow sensors were continuously recorded online using the HDAS hemodynamic data acquisition system (Department of Bioengineering, Universiteit Maastricht, Maastricht, The Netherlands).

Sixty µL/min of 0.9% saline were infused via the femoral vein. Two hours after completion of the instrumentation MAP, RVR and urine formation had stabilized, and rats were given 3 µg/kg nifedipine (*n* = 20) or vehicle (*n* = 18) intraperitoneally and, starting 1 h later, 1 µg/kg/min NPY or vehicle (*n* = 9–10) was infused for an additional 1 h. This period had been chosen because our previous studies had shown that NPY-induced alterations of diuresis and natriuresis require 45 min to fully develop [34]. MAP and RBF were measured every 5 min during the whole experiment and every minute during the first 10 min after the start of the NPY infusion. Urine was collected in pre-weighed tubes at 15 min intervals. At the end of the experiment, the rats were killed with an overdose of pentobarbital. Urine formation was quantitated gravimetrically assuming a specific gravity of 1.0, and samples were stored at 4 °C until analysis. Urinary sodium concentrations were determined with an Eppendorf flame photometer. Data on kaliuresis from this study have previously been reported as part of another publication [42].

### 4.2. Study II (NPY Bolus Injections)

Study II used similar methods as study I with some modifications [8,34]. Briefly, 24 male Wistar rats (weight 300–384 g) were allowed a 60 min recovery period during which 0.9% saline (60 μL/min) was infused. The fluid substitution was started immediately upon completion of the surgery. Three experimental groups (vehicle and 30 and 100 μg/kg/min nifedipine) of 8 rats each were tested. Findings from 1 rat in the vehicle group could not be evaluated for technical reasons bringing sample size to 7 rats for that group. After 60 min of equilibration, vehicle (0.1% ethanol in 0.9% saline) or 30 or 100 μg/kg/min nifedipine was infused for an additional 60 min. Thereafter, five consecutive bolus injections (0.3, 1, 3, 10 and 30 μg/kg NPY) were administered in a volume of 100 μL per 100 g body weight and injected within 30 s in 10 min intervals. During the experimental period MAP, HR and RBF were monitored every minute from 5 min before until 10 min after the bolus injection. The maximum response to NPY injection typically occurred within 1 min and recovered to baseline within 5 min. At the end of the experiment, the rats were killed with an overdose of anesthetic.

### 4.3. Chemicals

NPY was obtained from Saxon Biochemicals GmbH (Hannover, Germany), nifedipine from Sigma (Deisenhofen, Germany), thiobutabarbitone (Inactin^®^) from RBI (Natick, MA, USA), sodium pentobarbitone from Sanofi (Hannover, Germany), ketamine from Pittman-Moore GmbH (Burgwedel, Germany) and xylazine (Rompun^®^) from Bayer (Leverkusen, Germany). The test kit for creatinine measurement was from Boehringer-Mannheim (Mannheim, Germany). Nifedipine was dissolved at 1 mg/mL in ethanol and diluted with saline prior to use; special care was taken to avoid light exposure of nifedipine.

### 4.4. Data Analysis

Sample sizes had been defined before the experiments started. In the absence of knowledge of what a biologically relevant minimum difference for the various parameters would be, sample sizes were not based on power calculations but on our experience in previous studies on modulation of renal and vascular function by NPY [8,33,34,35,36,37]. RVR was calculated by dividing MAP by RBF [14] and expressed as mm Hg/(mL/min). The baseline in study I was defined as the average of the last 20 min before the start of NPY infusion for the hemodynamic parameters and of the last 30 min for the renal function parameters [33]. Baseline in study II was defined as the average of the hemodynamic parameters during the last 3 min before the first NPY injection [43]. Based on recent recommendations [44,45], group description data are expressed as means ± SD and effect sizes as means with 95% CI of the indicated number of animals. In line with the exploratory character of the study, no hypothesis-testing statistical analysis was performed.

## Figures and Tables

**Figure 1 molecules-26-04460-f001:**
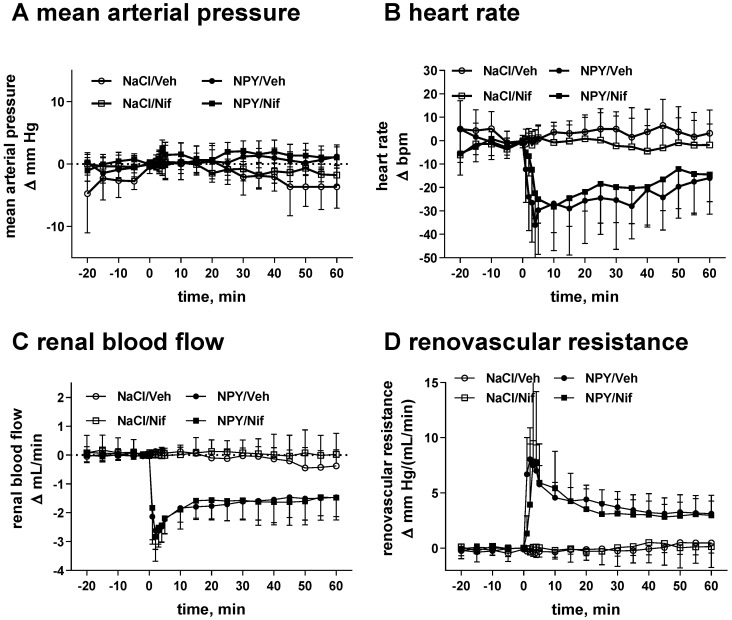
Effects of neuropeptide Y (NPY; from 060 min), its vehicle (NaCl), nifedipine (Nif) and its vehicle (Veh) on (**A**) mean arterial pressure, (**B**) heart rate, (**C**) renal blood flow, (**D**) renovascular resistance. Effects are means with 95% CI of 9 and 10 animals per group in the absence or presence of nifedipine, respectively.

**Figure 2 molecules-26-04460-f002:**
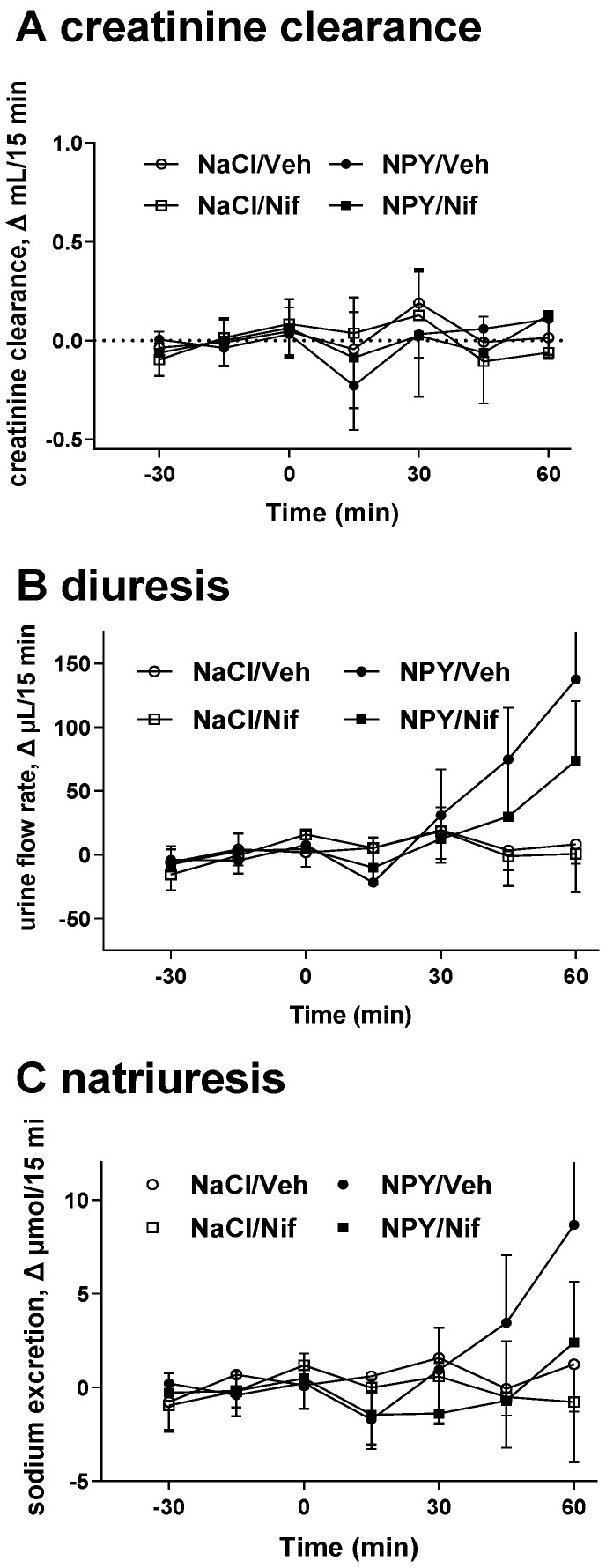
Effects of neuropeptide Y (NPY; from 0–60 min), its vehicle (NaCl), nifedipine (Nif) and its vehicle (Veh) on (**A**) creatinine clearance, (**B**) diuresis and (**C**) natriuresis. Effects are means with 95% CI of 9 and 10 animals per group in the absence or presence of nifedipine, respectively.

**Figure 3 molecules-26-04460-f003:**
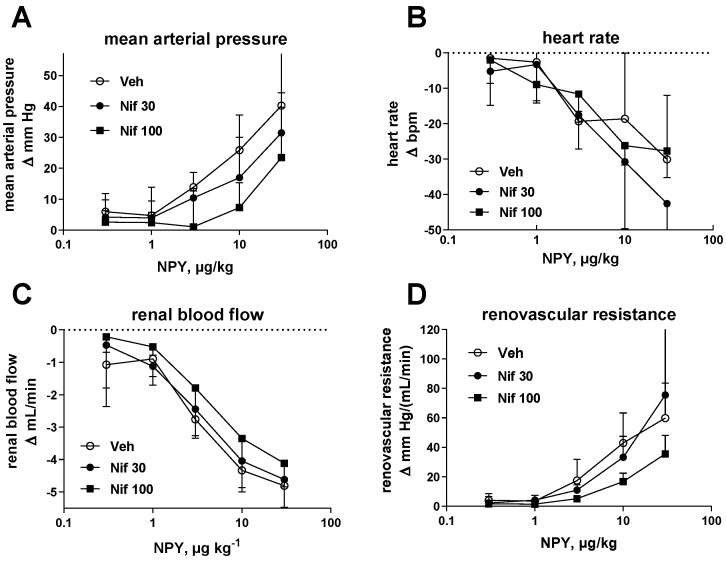
Effects of nifedipine on NPY-induced changes of cardiovascular parameters. Effects of vehicle (Veh) and nifedipine (30 and 100 μg/kg per min; Nif 30 and Nif 100, respectively) on changes of (**A**) MAP, (**B**) HR, (**C**) RBF and (**D**) RVR induced by bolus injections of NPY. Data are means ± SD of 7 control and 8 nifedipine-treated rats.

**Table 1 molecules-26-04460-t001:** Effects of nifedipine on basal values of mean arterial pressure (MAP, mm Hg), heart rate (HR, beats/min (bpm)), renal blood flow (RBF, mL/min), creatinine clearance (CCR, mL/15 min), diuresis (µL/15 min) and natriuresis (µmol/15 min) in study I. Data are shown as means ± SD for groups and as mean differences with their 95% confidence intervals based on 18 and 20 animals in the vehicle and nifedipine groups, respectively.

	Vehicle	Nifedipine	Difference
MAP	115 ± 13	121 ± 24	6 [−7; 17]
HR	321 ± 42	368 ± 38	47 [21; 73]
RBF	8.8 ± 1.7	7.9 ± 1.9	−0.9 [−2.1; 0.3]
CCR	1.2 ± 0.1	1.3 ± 0.7	0.1 [−1.5; 1.7]
Diuresis	123 ± 20	362 ± 0.148	0.177 [0.035; 0.319]
Natriuresis	1.2 ± 0.4	2.0 ± 0.1	0.8 [0.38; 1.22]

**Table 2 molecules-26-04460-t002:** Effects of 30 and 100 μg/kg/min nifedipine on basal values of mean arterial pressure (MAP), heart rate (HR) and renal blood flow (RBF) in study II. Data are shown as means ± SD based on 7, 8 and 8 animals, respectively, in the three groups. 95% CI for both nifedipine doses vs. vehicle included 0 for all parameters (data not shown).

	Vehicle	Nifedipine 30	Nifedipine 100
MAP	110 ± 4	106 ± 9	106 ± 20
HR	367 ± 24	328 ± 46	336 ± 35
RBF	5.6 ± 0.9	5.2 ± 0.9	5.3 ± 0.8

## Data Availability

All raw data can be found by object-linked embedding as Prism files in the figures.

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
