# Peer review of "Effects of Nifedipine on Renal and Cardiovascular Responses to Neuropeptide Y in Anesthetized Rats"

_molecules, 2021, doi:10.3390/molecules26154460_

Round 1

Reviewer 1 Report

The manuscript “Effects of nifedipine on renal and cardiovascular responses to neuropeptide Y in anesthetized rats” by Bischoff A et al. describes the effect of neuropeptide Y (NPY) and nifedipine on blood pressure, heart rate, renal blood flow, diuresis, natriuresis, and creatinine clearance. The authors conclude that Y5-mediated diuresis and natriuresis are more sensitive to inhibition by nifedipine than Y1-mediated renovascular effects.

I found the manuscript poor in several seccions. It lacks the rationale for several experimental conditions employed. Most important, the experimental conditions do not allow to reach the conclusion stated by the authors.  

The introduction misses relevant pieces of information. For instance, which is the renal distribution of Y5?. There are no hypothesis and/or aim for this paper. The title of the manuscript and the conclusion do not support each other.

What was the rational to use both, an infusion and bolus administration of NPY?

Why the rats were  nephrectomized? No explanation was provided. Furthermore, no discussion on the characteristics of the experimental model was included.

Due to the null employment of specific Y1-, Y5- antagonists, no categorical assumption of their participation can be stated.

How the authors exclude the participation of angiotensin II mediating NPY –induced effects?  

No statistical significance is denoted along the results. Therefore, no differences can be claimed.

What did you mean in lines 84 and 85?

According to the first paragraph of the discussion, the findings of the manuscript confirmed previous reports. Then what is new?

I disagree with your statement: “… NPY treatment with nifedipine attenuated the diuretic any natriuretic responses, indicating that Y5 receptors may also use L-type Ca2+ channels…” It would be necessary to perform the experiments blocking Y5 receptor. The same applies for the following sentence:  “… present data provide the first evidence that Y5 NPY receptor-mediated effects of diuresis and natriuresis are attenuated by the L-type Ca2+ channel inhibitor nifedipine”.

There is a mistyping writting the strain of the rats. It should say Wistar, not Wister.

Why the rats for study I and II were of different weight?

It is remarkable that author’s self-reference reaches 34%!

Author Response

“The manuscript “Effects of nifedipine on renal and cardiovascular responses to neuropeptide Y in anesthetized rats” by Bischoff A et al. describes the effect of neuropeptide Y (NPY) and nifedipine on blood pressure, heart rate, renal blood flow, diuresis, natriuresis, and creatinine clearance. The authors conclude that Y5-mediated diuresis and natriuresis are more sensitive to inhibition by nifedipine than Y1-mediated renovascular effects.

I found the manuscript poor in several seccions. It lacks the rationale for several experimental conditions employed. Most important, the experimental conditions do not allow to reach the conclusion stated by the authors.”

We thank the reviewer for commenting on our manuscript. As can be seen from our specific replies below, we feel that his/her conclusion that “the experimental conditions do not allow to reach the conclusions stated by the authors” is largely based on misunderstandings.

“The introduction misses relevant pieces of information. For instance, which is the renal distribution of Y5?. There are no hypothesis and/or aim for this paper. The title of the manuscript and the conclusion do not support each other.”

To the best of our knowledge, no data on the distribution of Y5 receptors in the kidney, including rat kidney, have been reported. The only data we are aware of are based on NPY binding without attempts to separate subtypes. Therefore, this part of the comment cannot be addressed.

No hypothesis was stated because the underlying studies were not designed to be hypothesis-testing but to be exploratory. In this regard the aim of the two studies (exploring effects of NPY) has clearly been stated in the Abstract (l. 14-15) and the main Introduction (l. 50-54).

We do not understand the second part of the comment (title and conclusion do not support each other). What we observed and reported are renal and cardiovascular responses; whether we can directly attribute them to specific sites e.g., within the kidney is not directly relevant to our conclusions.

“What was the rational to use both, an infusion and bolus administration of NPY?”

Based on the time necessary to observe alterations of diuresis and natriuresis (see Figure 2, which is consistent with several of our previous studies) an infusion design is required to study the tubular effects. On the other hand, as shown in Figure 1 of the manuscript and consistent with observations in several previous studies, renovascular effects attenuate within a few minutes after reaching an initial peak. Based on our previous studies (reference #11 in original submission, now #12), this involves AT1 receptors. Based on this attenuation over time, the infusion design is suboptimal for studying the vascular effects. Therefore, we used the bolus design to study the vascular responses and their modification by nifedipine in more detail. This had been explained in l. 133-135 of the original submission (now line 138-140). Moreover, only the bolus design can easily test multiple NPY doses within one animal, which is necessary to study dose-response relationships and their possible shifts by nifedipine. The latter aspect is explained in l. 140-141 of the revised manuscript.

“Why the rats were  nephrectomized? No explanation was provided. Furthermore, no discussion on the characteristics of the experimental model was included.”

Unilateral nephrectomy several days prior to the experiment is a standard approach in this field. It allows specific manipulation of one kidney (e.g., intrarenal administration, acute denervation, decapsulation) without interference in outcomes from the other kidney. Since we had performed such interventions in several of our earlier studies, we kept it to have a consistent study design and facilitate comparison of data across studies.

Our basic findings on NPY are fully consistent with those in many previous studies (l. 103-106 of original submission, now l. 105-108). Therefore, we did not wish to repeat a discussion of the overall model that we had provided extensively in our previous papers. If the referee and editor so wish, we will be happy to add this type of discussion at short notice.

“Due to the null employment of specific Y1-, Y5- antagonists, no categorical assumption of their participation can be stated.”

We respectfully disagree with this statement. We and others have extensively characterized the NPY receptor subtypes mediated the renal and vascular responses studied here based on various subtype-selective agonists and antagonists in previous studies. This was described in l. 107-114 of the original submission (now l. 109-116). We do not see a need to repeat the subtype characterization in every follow-up study.

“How the authors exclude the participation of angiotensin II mediating NPY –induced effects?”  

In a previous study (reference #11 of original submission) we had studied effects of losartan (no effect on tubular NPY responses) and ramipril (enhanced diuresis), but the latter could be attributed to the ACE inhibitor effect on bradykinin as shown by additional experiments with icatibant. The lack of involvement of angiotensin II is now discussed (l. 114-116).

“No statistical significance is denoted along the results. Therefore, no differences can be claimed.”

The senior author of the manuscript (MCM) teaches statistics courses at several universities. As already discussed in the original submission (l. 224-226, now l. 230-233), recent recommendations from professional statisticians recommend applying hypothesis-testing statistics to very limited settings and to rather report effect sizes with their confidence intervals. For instance, an entire issue of the journal of the American statistical society was dedicated to this. As the originally quoted reference includes MCM as the author (but also two professional statisticians), we have added a reference in this regard from other statisticians published in Nature and co-signed by >800 leading other statisticians.

“What did you mean in lines 84 and 85?”

We find this text very clear: there was a numerical reduction, but its confidence interval included 0 (the equivalent of p > 0.05) and it was not dose-dependent.

“According to the first paragraph of the discussion, the findings of the manuscript confirmed previous reports. Then what is new?”

There may be a misunderstanding. What we say in the 1st paragraph of the discussion is that the observed effects of NPY (in the absence of nifedipine) were in line with several previous studies. What is new, obviously is that the previous studies have not tested effects of nifedipine or any other L-type Ca channel blocker.

“I disagree with your statement: “… NPY treatment with nifedipine attenuated the diuretic any natriuretic responses, indicating that Y5 receptors may also use L-type Ca2+ channels…” It would be necessary to perform the experiments blocking Y5 receptor. The same applies for the following sentence:  “… present data provide the first evidence that Y5 NPY receptor-mediated effects of diuresis and natriuresis are attenuated by the L-type Ca2+ channel inhibitor nifedipine”.”

We are happy to acknowledge that the referee does not share our interpretation. However, as pointed out above, studies with several subtype-selective agonists and antagonists had established previously that the tubular NPY responses in rats are mediated by Y5 receptors. Unless the referee questions the validity of those previous studies, we do not see what would invalidate our interpretation of the present observations. Apparently, the editor also disagreed with this specific reviewer comment as we were granted only 5 days to submit a revised manuscript, which implies that no additional data are required according to the editor.

“There is a mistyping writting the strain of the rats. It should say Wistar, not Wister.”

This typo has been corrected.

“Why the rats for study I and II were of different weight?”

Rats in study I weighed 298-415 g, those in study II 300-384 g, i.e., all values from study II were within the range of study I. We find this variability in weight between rats normal.

“It is remarkable that author’s self-reference reaches 34%!”

We did not intend to include inappropriate self-citations. However, after an initial study by Smyth using isolated perfused kidneys, our lab has been the pioneer of studying renal responses to NPY. This makes it unavoidable to cite many of those papers.

Reviewer 2 Report

This is a very nice paper describing the L-type Ca2+ channel inhibitor nifedipine to probe the involvement of Neuropeptide Y (NPY) in the regulation of diuresis and natriuresis. The study design was straightforward and clear. The paper was very nicely written with an appropriate interpretation of the data.  Please consider the following minor comments:

Comments

  1. Authors comment that the present data do not allow definitive conclusions whether the involvement of L-type Ca2+ channels reflects a direct coupling of Y5 receptors; alternatively, it may reflect that receptors activated by cyclooxygenase products may lead to activation of such channels. How can authors provide the relation of nifedipine on cyclooxygenase products like indomethacin on renal tubular NPY responses?
  2. It is preferable if authors can provide the in vitro studies, at least on renal tubular cells preferentially testing Y5 receptors to address their hypothesis.

Author Response

“This is a very nice paper describing the L-type Ca2+ channel inhibitor nifedipine to probe the involvement of Neuropeptide Y (NPY) in the regulation of diuresis and natriuresis. The study design was straightforward and clear. The paper was very nicely written with an appropriate interpretation of the data.  Please consider the following minor comments:”

We thank the reviewer for his/her encouraging comments.

Comments

  1. “Authors comment that the present data do not allow definitive conclusions whether the involvement of L-type Ca2+ channels reflects a direct coupling of Y5 receptors; alternatively, it may reflect that receptors activated by cyclooxygenase products may lead to activation of such channels. How can authors provide the relation of nifedipine on cyclooxygenase products like indomethacin on renal tubular NPY responses?”

We have no data supporting the view that nifedipine acts on cyclooxygenase products. That sentence had only been introduced as a note of caution: if the effect of NPY on tubular function involves the intermediary formation of cyclooxygenase products, we cannot exclude on theoretical grounds that nifedipine interferes with the latter process and not directly with the signaling of NPY receptors.

  1. “It is preferable if authors can provide the in vitro studies, at least on renal tubular cells preferentially testing Y5 receptors to address their hypothesis.”

This is a very good proposal and would also potentially address the above question whether nifedipine acts at the level of NPY receptors or is involved in the signaling pathway of cyclooxygenase products. However, we have never worked with isolated tubules. As the editor has given us only 5 days to submit a revision, we interpret this as a decision that additional experiments are not needed at this time.

Reviewer 3 Report

Thank you for the possibility to review this paper.

Chech carrefully the reference sequence. I.e. in Introduction you stared with reference 18  and 20 (line 30).

Usually confidence intervals are expressed in this way x (xx-xy): in line 59 47 [21; 73] -> 47 [21-73].

Please insert a legend in table 1 to explain HR, MAP, RBF etc.... I suggest to clarify if exits a statistical difference between veichle and nifedipine. Check the normality of the data wiht a Shapiro test or Kolgogorov test and apply the correct statistical test form means. If you use a mean and standard deviation I suppuse that the data are normally distribuited. in this case you can use a ttest. on the contrary explain your data in median and interquartile range and use a wilkoxon test. in any case, check carrefully this statistical aspects.

in line 217 you explain the reason why you d not perform a sample size  calculation but you refered to yours previous studies on the argument. the sample size reflect this aspect. if you have an idea of the expeted changes you can perform a sample size calculation. to avoid this misunderstand, i suggest to added "pilot study" in this section and, if you agree, in the title.

in the discussion section it would be interesting added a short paragraph  about nifedipine effects in the humans and the potential traslation in humans of your research.

Author Response

“Chech carrefully the reference sequence. I.e. in Introduction you stared with reference 18  and 20 (line 30).”

Starting with reference 18 and 20 is due to the fact that author instructions of this journal ask to list references in alphabetical order, not by sequence of being cited.

“Usually confidence intervals are expressed in this way x (xx-xy): in line 59 47 [21; 73] -> 47 [21-73].”

In our experience, both ways of showing confidence intervals are found in the literature. The reason we had chosen for the [xx; yy] option is that confidence intervals including 0 (e.g., table 1) can become confusing when expressed as [xx-yy]; for instance, [-7; 17] makes it clear that the intervals span 0, whereas [-7-17] can be confusing.

“Please insert a legend in table 1 to explain HR, MAP, RBF etc.... I suggest to clarify if exits a statistical difference between veichle and nifedipine. Check the normality of the data wiht a Shapiro test or Kolgogorov test and apply the correct statistical test form means. If you use a mean and standard deviation I suppuse that the data are normally distribuited. in this case you can use a ttest. on the contrary explain your data in median and interquartile range and use a wilkoxon test. in any case, check carrefully this statistical aspects.”

We have added explanations of the abbreviations to the legend of table 1 (l. 61-62) and also that of table 2 (l. 94-95).

The senior author of the manuscript (MCM) teaches statistics courses at several universities. As already discussed in the original submission (l. 224-227, now l. 230-233), recent recommendations from professional statisticians recommend applying hypothesis-testing statistics to very limited settings and to rather report effect sizes with their confidence intervals. For instance, an entire issue of the journal of the American statistical society was dedicated to this. As the originally quoted reference includes MCM as the author (but also two professional statisticians), we have added a reference in this regard from other statisticians published in Nature and co-signed by >800 leading other statisticians. Therefore, we had intentionally decided not to apply hypothesis-testing statistical tests at all, whether parametric or not.

The question whether a parameter exhibits a normal distribution or not, is indeed relevant. However, the question of normality relates to the underlying populations, not the samples at hand. We feel that the current sample sizes are too small to make robust inference on a normal distribution of the underlying population. However, the entire field (including the data from large clinical studies) reports blood pressure and heart rate as means ± SD, assuming a normal distribution.

By the way: the Komogorov-Sminov test had historical value but is considered as outdated by most statistician. If one wanted to test for normality, we feel that the D’Agostino & Pearson omnibus K2 test is most appropriate. However, this is not a relevant question in our view for the above reasons.

“in line 217 you explain the reason why you d not perform a sample size  calculation but you refered to yours previous studies on the argument. the sample size reflect this aspect. if you have an idea of the expeted changes you can perform a sample size calculation. to avoid this misunderstand, i suggest to added "pilot study" in this section and, if you agree, in the title.”

We agree that sample size calculations for the effects of NPY would have been feasible based on the experience from our previous studies. However, we had no prior knowledge on expected effect sizes for nifedipine. Thus, we fully agree that the present experiments were not designed to be hypothesis-testing as this would require prior sample size calculations. Rather we consider our experiments to be exploratory, as stated in l. 226-227 of the original submission (now l. 232-233). Thus, sample sizes were fully in line with our previous studies, whereas we would consider it a pilot study if they had been smaller.

“in the discussion section it would be interesting added a short paragraph  about nifedipine effects in the humans and the potential traslation in humans of your research.”

That is an interesting suggestion. The effects of nifedipine as a blood pressure-lowering agent in humans are textbook knowledge. While effects of nifedipine against renal effects of NPY have not been studied in humans, we added a statement and reference on the natriuretic effect of nifedipine in patients, which is in line with our data from table 1. We have added wording and a representative reference to this effect on l. 124-125.

Reviewer 4 Report

In this manuscript Bischoff et al. shows that nifedipine attenuated Y5 NPY receptor-mediated effects of diuresis and natriuresis in rats received either continuous infusion or bolus injections of NPY. The experimental approach and methods used appear clean and technically sound.

1) The role of neuropeptide Y1 and Y5 in the regulation of cardiovascular and renal effects is unclear. I would suggest that the authors to include a schematic illustrating the signal transduction mechanisms of L-type Ca2+ channel blocker nifedipine in the regulation of diuresis and natriuresis by NPY-5 receptor.

2) Have the authors consider looking at another calcium channel blocker, Nisoldipine, which is structurally similar to nifedipine, to determine diuretic and natriuretic effects for NPY-5 and compared with that of nifedipine?

3) In this study, the authors conclude that Y5-mediated diuresis and natriuresis are more sensitive to inhibition by nifedipine than Y1-mediated renovascular effects, but it is unclear to me whether or not this is a good thing? And if there is a direct relevance to clinical effects in human studies? Please clarify.

Author Response

“In this manuscript Bischoff et al. shows that nifedipine attenuated Y5 NPY receptor-mediated effects of diuresis and natriuresis in rats received either continuous infusion or bolus injections of NPY. The experimental approach and methods used appear clean and technically sound.”

Thank you for your kind comments.

  • “The role of neuropeptide Y1 and Y5 in the regulation of cardiovascular and renal effects is unclear. I would suggest that the authors to include a schematic illustrating the signal transduction mechanisms of L-type Ca2+ channel blocker nifedipine in the regulation of diuresis and natriuresis by NPY-5 receptor.”

This is an interesting suggestion that we had entertained during the writing of the original submission. We had chosen not to do so because we know that the tubular effects occur via Y5 receptors, but do not know where these are located and found that they involve the intermediary formation of cyclooxygenase products (s. l. 124-128 of original submission, now l. 130-133). Thus, any schematic diagram would be highly speculative. We prefer not to engage in too much speculation, particularly because referee #1 already felt that we had too much of it in the original submission.

  • “Have the authors consider looking at another calcium channel blocker, Nisoldipine, which is structurally similar to nifedipine, to determine diuretic and natriuretic effects for NPY-5 and compared with that of nifedipine?”

We have not considered comparing nifedipine to other Ca2+ channel blockers. To the best of our knowledge, this is the first study showing effects of any such blocker. In humans, diuretic effects of such drugs apply to the entire class.

  • “In this study, the authors conclude that Y5-mediated diuresis and natriuresis are more sensitive to inhibition by nifedipine than Y1-mediated renovascular effects, but it is unclear to me whether or not this is a good thing? And if there is a direct relevance to clinical effects in human studies? Please clarify”

Another good question. Unfortunately, little is known on the role of NPY in the regulation of renal function. Given that the expression of NPY receptors differs a lot between species (highest in rabbits, very low to non-detectable in humans) we find it difficult to speculate how nifedipine would interact with NPY in humans and whether this would be a good or bad thing. The overall utility of Ca2+ channel blockers in clinical medicine is well established, and the current data are unsuitable to cast doubt on that role.

Round 2

Reviewer 1 Report

… we feel that his/her conclusion that “the experimental conditions do not allow to reach the conclusions stated by the authors” is largely based on misunderstandings

Response: As authors, it is our due to clearly state the information we report in order to avoid misunderstandings.  

No hypothesis was stated because the underlying studies were not designed to be hypothesis-testing but to be exploratory.

Response: …………. I checked on both mentioning and it is stated in the abstract: “… we explored whether nifedipine affects NPY-induced diuresis and natriuresis. However, in the introduction it says: “… the present studies have used the L-type Ca2+ channel inhibitor nifedipine to probe the involvement of such channels in the regulation of diuresis and natriuresis representing Y5 receptors…”

Authors: We do not understand the second part of the comment (title and conclusion do not support each other).

Response: Following practical rules on the title of a manuscript, it is known that a good idea is to use the title that accurately describes the contents of your manuscript. Since this reviewer sees that NPY and Y5 receptors carry a more relevant role in the research than nifedipine (used as a tool to probe the involvement of L-type Ca2+ channels), I would consider a slightly different title. See the title “Effects of nifedipine on renal and cardiovascular responses to neuropeptide Y in anesthetized rats” and the conclusion: “We conclude that Y5-mediated diuresis and natriuresis are more sensitive to inhibition by nifedipine than Y1-mediated renovascular effects.” It was an observation.

Authors: Based on the time necessary to… This had been explained in l. 133-135 of the original submission (now line 138-140). Moreover, only the bolus design …. is explained l. 140-141 of the revised manuscript.

Response: It would be highly appreciated by a reader of your work, to have the information in the same paper. That would help to understand and make the analysis fluent, without the inconvenience (in some cases impossibility) to read several other papers looking for information that you can provide with no effort.

On the other hand, I double checked current version of the manuscript. Lines 138-140 do not mention the point, neither lines 140-141 made reference to the rational for using both, an infusion and bolus administration of NPY.

Authors: Unilateral nephrectomy several days prior to the experiment is a standard approach in this field.

Response: According to Guide for the Care and Use of Laboratory Animals: Eighth Edition (2011), “Multiple major surgical procedures on a single animal are acceptable only if they are (1) included in and essential components of a single research project or protocol, (2) scientifically justified by the investigator, or (3) necessary for clinical reasons.” From the explanation you provided, I do not see a justification to have nephrectomized the rats. In any case, the rational must be provided in the manuscript.    

(National Research Council 2011. Guide for the Care and Use of Laboratory Animals: Eighth Edition. Washington, DC: The National Academies Press. https://doi.org/10.17226/12910.).   

Regarding the discussion on the characteristics of the experimental model. The bottom-line question was: How the nephrectomy modifies the experimental model? , Is the lack of one kidney changing the renovascular or cardiovascular response?  

Authors: We respectfully disagree with this statement….

Response: The paper you refer to: Receptor subtypes Y1 and Y5 are involved in the renal effects of neuropeptide Y.  Angela Bischoff, Prodromos Avramidis, Wilhelm Erdbrügger, Klaus Münter, Martin C. Michel. First published: 17 February 2009 https://doi.org/10.1038/sj.bjp.0701028 concludes that “the present study demonstrates that NPY reduced RBF via a classical Y1 receptor. In contrast enhancements of diuresis, natriuresis, and calcineuresis may occur by a similar but distinct receptor which  is activated by PYY3-36 and not inhibited by BIBP3226… “ There is no straightforward mention to Y5 receptor. Currently there is NPY Y5 receptor antagonist (NPY 5RA972, available at Tocris).

Authors: The senior author of the manuscript (MCM) teaches statistics courses at several universities….

Response: 1.- The lines you refer to do not mention anything related to statistical mater or the paper you refer to, 2.- Assuming that the new way to report the results is correct, how do you manage the fact the SD and CI overlap among the groups? Does the case allow you to categorically conclude “differences”?

Authors: There may be a misunderstanding….

I suggest to rephrase the paragraph and use those references to highlight the results and conclusions you reached with this work. Otherwise it is confusing and seems that you just reproduced what is already published.

Authors: We are happy to acknowledge that the referee does not share our interpretation…. Apparently, the editor also disagreed with this specific comment as we were granted only 5 days to submit a revised manuscript, which implies that no additional data are required according to the editor.

Response: Following the references you provided, I have not been able to find the report with experiments that categorically support your claims. On the other hand, we are not to judge Editor’s decisions.